# Adaptive oscillators support Bayesian prediction in temporal processing

**Keith B. Doelling** [1,2]*, **Luc H. Arnal**[1‡], **M. Florencia Assaneo**[3‡]*

**1** Institut Pasteur, Université Paris Cité, Inserm UA06, Institut de l'Audition, Paris, France, **2** Center for Language Music and Emotion, New York University, New York, New York, United States of America, **3** Instituto de Neurobiología, Universidad Nacional Autónoma de México, Santiago de Querétaro, México

‡ These authors jointly supervised this work.
* keith.doelling@pasteur.fr (KBD); fassaneo@inb.unam.mx (MFA)

**Data Availability Statement:** All relevant data can be found in the paper's Supporting Information files.

## Abstract

Humans excel at predictively synchronizing their behavior with external rhythms, as in dance or music performance. The neural processes underlying rhythmic inferences are debated: whether predictive perception relies on high-level generative models or whether it can readily be implemented locally by hard-coded intrinsic oscillators synchronizing to rhythmic input remains unclear and different underlying computational mechanisms have been proposed. Here we explore human perception for tone sequences with some temporal regularity at varying rates, but with considerable variability. Next, using a dynamical systems perspective, we successfully model the participants behavior using an adaptive frequency oscillator which adjusts its spontaneous frequency based on the rate of stimuli. This model better reflects human behavior than a canonical nonlinear oscillator and a predictive ramping model–both widely used for temporal estimation and prediction–and demonstrate that the classical distinction between absolute and relative computational mechanisms can be unified under this framework. In addition, we show that neural oscillators may constitute hard-coded physiological priors–in a Bayesian sense–that reduce temporal uncertainty and facilitate the predictive processing of noisy rhythms. Together, the results show that adaptive oscillators provide an elegant and biologically plausible means to subserve rhythmic inference, reconciling previously incompatible frameworks for temporal inferential processes.

## Author summary

Real-life sequence processing requires prediction in uncertain contexts. Temporal predictions are thought to be manifest by a dual timing mechanism: a relative timing mechanism measuring timing relative to an expected rhythm, and an absolute timing mechanism which measures absolute interval differences. We show that these mechanisms can be unified into a Bayesian Framework, reformulating the two components into the combination of prior and sensory measurement, and demonstrate that this framework can be implemented by an adaptive neural oscillator.

**Funding:** Support was provided by UNAM-DGAPA-PAPIIT IA200223 (MFA), IBRO Return Home fellowship (MFA), Fondation Fyssen Postdoctoral Grant (KBD) and Fondation pour l'Audition grant RD-2020-10 (LHA). The funders had no role in study design, data collection and analysis, decision to publish, or preparation of the manuscript.

**Competing interests:** The authors have declared that no competing interests exist.

## Introduction

Tracking the rhythm of a sequence of events–as in speech or music–helps the brain to reduce temporal uncertainty, facilitating the processing of upcoming events [1]. Many empirical works in this domain have focused on the potential role of neural oscillations as a neurophysiological substrate for predictions in the time domain [2–5]. In this view, neural oscillators synchronize their excitability phase with external sequences, thereby reducing internal noise and optimizing the processing of incoming events [6–8]. In this sense, the phase of a neural oscillation can be used as an index for prediction in time, a mechanism that may be considered as constitutive of the inferential process.

One issue that arises in adjudicating between underlying models in rhythm perception is that all reasonable models will yield the same prediction under *perfect rhythmicity*. However, natural stimuli generally possess regular temporal statistics often construed as pseudo-rhythmicity: ecological sounds are rarely perfectly rhythmic and instead maintain variability in their timing [9–11]. Using temporally variable sequences is arguably necessary to capture the essence of internal models, namely the extraction of noisy patterns to reduce uncertainty about future events. As such, by introducing temporal uncertainty–i.e. jittering the events–in the sequence, we can better distinguish between models: variance in the intervals leads to variance of predictions in the models.

Variance in timing also arguably represents a challenge for the neural oscillatory synchronization hypothesis [12]. Such mechanisms could become less efficient (or even detrimental) in the context of more irregular temporal patterns. Recently, we theoretically demonstrated that even in the face of such variability a basic oscillator can still synchronize to a variable rhythm [13]. Does this kind of synchronization still work as a mechanism for human temporal prediction without perfect isochronicity? Can the phase adjustments that lead to successful synchronization be reconciled with a Bayesian account of perception? To answer these questions, we devised temporal sequences with regular statistics but with clear temporal jitter. Participants therefore could predict when the next tone was most likely to occur without being able to have total confidence. This protocol allowed us to determine how temporal expectations are executed under these adverse—but more naturalistic—conditions.

While previous studies have used similar experimental designs (see for example, [14]), the computational mechanism driving the subject's responses are often taken for granted and the analyses were conducted according to their assumptions. In this line, two distinct computational mechanisms have been proposed in the literature to subserve perceptual timing: absolute or relative timing [15]. Absolute timing relies on the estimation of the concrete duration of discrete time events. Relative timing, instead, refers to the computation of a time interval with respect to a standard defined by a temporal regularity present in the environment. Typically, it is assumed that one or the other time assessment drives behavior according to the specificities of the experimental design. For example, it has been hypothesized that timing of intervals in irregular time sequences recruit absolute timing mechanisms while regular sequences recruit relative timing ones [16]. Here, in contrast with previous studies, we do not presuppose that one or another timing mechanism will drive participants' expectations. As an alternative, we adopt a data driven approach, where the two classical timing computational mechanisms emerge as opposite poles of participant behavior that range from one to the other. By taking this feature into account, we develop a more complete understanding of human behavior.

Our findings show that what seems to be two different timing mechanisms (i.e., relative, or absolute timing) can be different sides of the same coin. While participants' responses reflected different algorithms (i.e., relative, or absolute) subserving perceptual timing depending on the

temporal features of the acoustic stimulus, we show that the whole pattern of responses can be captured by a single biophysical neural model. More precisely, we propose an adaptive frequency oscillator as a reasonable candidate for temporal prediction under these more ambiguous circumstances by taking qualities typically used in other models of the literature (e.g., predictive ramp) to improve performance of the classic oscillator model. In addition, we show that participants' behavior is consistent with Bayes' Theory with regards to temporal prediction in sequences. Taken together, we show the advantage for proponents of oscillatory models to develop more complexity in their theories to accommodate realistic scenarios of human perception.

## Results

### Human behavior, absolute vs relative perceptual timing

We evaluated human perception for pseudo-rhythmic sequences of tones. Participants completed 150 trials. On each one they listened to 9 to 11 tones, and they indicated whether the last tone was advanced or delayed. Each tone of the sequence was jittered from a perfectly rhythmic underlying structure. A gaussian (all tones except for the last one) or a uniform random jitter (last tone) was applied to each tone position (see Fig 1A and Methods). With this design we evaluated three different cohorts of participants, assigning different underlying rhythmic structures for each cohort: 1.2 Hz, 2Hz or 4Hz rhythms.

As stated in the introduction, two different computational mechanisms have been proposed in the literature accounting for perceptual timing: absolute and relative timing. These two mechanisms differ largely in terms of how jitter accumulates. The absolute method averages the absolute intervals between preceding tones and adds them to the timing of the current tone to yield an expectation of the next tone. On the other hand, the relative method infers an isochronous rhythm across the preceding intervals and estimates the next tone as the next iteration of this rhythm plus noise. As such, the absolute method has stronger expectations for drift as predictions more readily incorporate the jitter of each tone. Here, we explored which of these two classically proposed perceptual timing mechanisms drives participants behavior. We generated a series of stimuli jittering the tone times in a method consistent with the relative mechanism as we are interested especially in rhythmic temporal prediction. Crucially, gaussian noise was added to the perfect isochronous structure making the two computational mechanisms of perceptual timing (i.e., absolute, and relative) to grant different predictions on when the next tone in the sequence will take place (see Figs 1AB and S1). Participants listened to the sequences and for each one estimated if the final tone was early or later than they expected. Sigmoids were fitted to the subjects' responses (i.e., 1 = late, 0 = early) as a function of the position of the last tone (i.e., probe) with reference to the expected timing for each of the two mechanisms: absolute or relative timing (see Fig 1A and 1B and Methods). A comparison between the slopes of the fittings allowed us to identify the computational mechanism that better explains the participants' behavior (see Fig 1C). Following this procedure, we analyzed participants' responses across the three distinct stimulus rates used in our experiment (1.2 Hz, 2Hz or 4Hz). As shown in Fig 2A, stimulus rate significantly affected which computational mechanism better adjusts perceptual timing. At 1.2 Hz, participants' responses were better explained by absolute timing, showing higher (steeper) slope values for the sigmoids fitted according to absolute timing compared to the ones adjusted to the relative timing predictions (Wilcoxon signed-rank test, two-sided $p < 0.001$). On the other hand, participants in the 2 Hz conditions showed significantly greater alignment with relative timing (Wilcoxon signed-rank test, two-sided 2 Hz: $p = 0.022$), while those in the 4Hz conditions also aligned more heavily with relative timing (i.e., higher slope values for the relative timing sigmoids compared to the

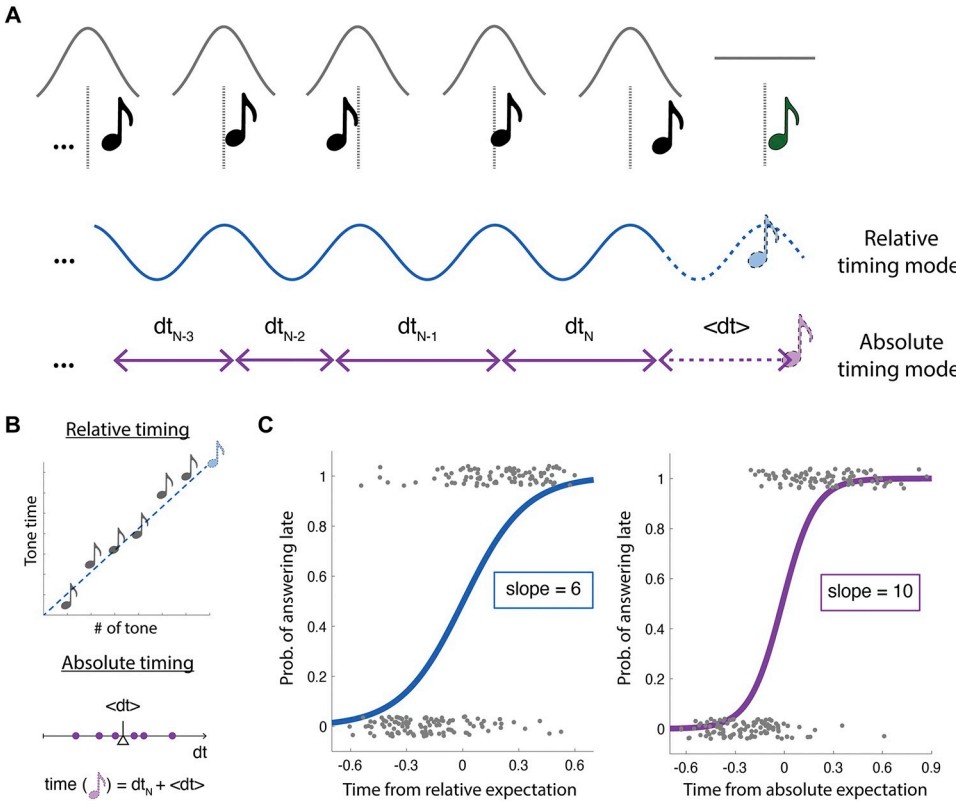

**Fig 1. Experimental design and analysis framework. A.** Schema reflecting the trials design and the different plausible perceptual timing computations used to estimate the expected final location. The position of the tones is jittered from an isochronous underlying structure (gray vertical lines) according to different probability functions (gray traces): a gaussian distribution was chosen for the cue tones (black) and a uniform distribution for the final probe tone (green). According to relative perceptual timing (blue), expectation reconstructs the underlying isochronous structure. According to the absolute perceptual timing (magenta), predictions rely on the estimation of the mean absolute duration of the intervals. **B**. Computations used to calculate relative and absolute estimations. Relative timing uses linear regression, fitting a line to predict timing of the next tone based on its position in the sequence. Absolute stores the intervals between tones, averages and adds this interval to the final tone location. **C.** Participants' psychometric functions were fitted according to the two perceptual timing frames. Analysis of one exemplar participant. Responses (coded 0 for early, 1 for late) are compared to both the relative (left) and the absolute computations (right). Logistic model is fitted to the results and the slope of the fit is used as a measure of consistency with the given computational mechanism. This example shows a participant more consistent with the absolute computation of time. Dots represent single trial answers with a small vertical jitter added for visualization purposes. The data set used to generate these two panels can be found in S1 Data. The music note icons is used from openclipart.org.

absolute timing ones, and 4 Hz: $p < 0.001$). Furthermore, slope differences (i.e., $Slope_{ABS} - Slope_{REL}$) significantly differed between studies (Mann-Whitney-Wilcoxon test, two sided, Bonferroni corrected for 3 comparisons: $p_{1.2Hz-2Hz} < 0.001$, $p_{1.2Hz-4Hz} < 0.001$, $p_{2Hz-4Hz} = 0.028$).

Next, we sought to understand if this effect is truly the result of a shift in frequency or is instead related by a third variable. To do so, we first plotted the slopes of the logistic fits of each model against each other across all conditions (see Fig 2B). This analysis shows qualitatively that results from each experiment are unified in their relationship between the two timing mechanisms, biased towards relative timing with worse overall performance and towards absolute timing with better overall performance. To clarify and quantify this result, in Fig 2C we rotated the axes of Fig 2B comparing the overall performance of the models ($slope_{ABS} + slope_{REL}$) to the difference between the two models ($slope_{ABS} - slope_{REL}$). Note that this transformation represents a linear rotation of the raw data to enable more intuitive analysis. To

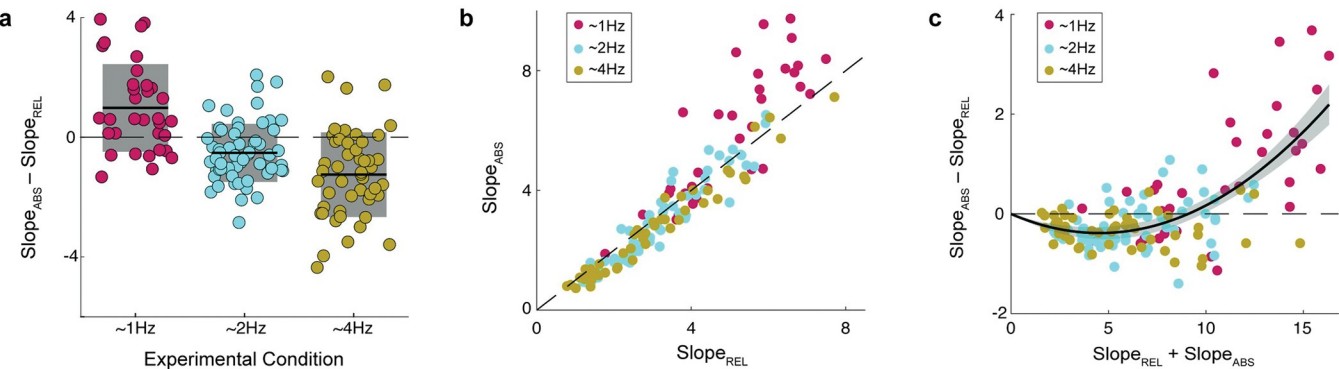

**Fig 2. Comparisons between the two perceptual timing mechanisms used to fit participants' responses. A.** Difference of the slopes obtained by fitting a logistic regression for the participants answers as a function of the probe time computed from: the expected time according to the absolute (Slope$_{ABS}$), or the expected time according to the relative (Slope$_{REL}$) computational mechanisms (N$_{1Hz}$ = 35, N$_{2Hz}$ = 63 and N$_{4Hz}$ = 51). Black line indicates the mean value and shaded region the standard deviation. **B.** Fit of relative vs absolute mechanism for each participant across three experiments. Dashed line indicates the identity line, where each mechanism would perform equally well. **C.** Perceptual timing mechanism alignment as a function of the overall performance. Slope difference between the two normalized logistic regressions as a function of the summed slope across algorithms. The black line represents a second degree polynomial fit and the shaded region the 95% Prediction Interval. The parameters of the fitted polynomial are: $y = ax^2+bx$ with $a = 0.0196$, $b = -0.175$. Data for Fig 2A,B & C can be found in S2 Data.

quantify the function that drives the relationship between performance and algorithm dominance, we fit the dataset across experiments with polynomial functions of 1, 2, and 3 degrees. We compared the models using the Akaike Information Criterion (AIC) and found the second-order polynomial fit to be best (AIC, first: 334.30, second: 249.62, third: 250.07). The parameter fit confirms that for overall low performance (i.e., slope$_{ABS}$ + slope$_{REL}$ < 8.12) human behavior significantly aligns to relative timing (Fig 2C). Furthermore, the model fits the data remarkably well (adj. R-square = 0.71).

## Adaptive frequency model

In the previous section we explored perceptual timing, according to the two classical computational mechanisms proposed in the literature: absolute and relative timing. We found that human perception better aligns to relative timing for higher frequencies, where performance is low, and to absolute timing for lower frequencies, where performance increases. Relative and absolute timing are computational algorithms constructed by researchers based on behavioral observations. There is not a straight relationship between these algorithms and an underlying neural mechanism. In this section we sought to reconcile the behavioral observations described in terms of the two classically proposed timing mechanisms (i.e., absolute, and relative timing) with a plausible biophysical model. More precisely, we explored whether a single neural model could explain the switch from relative to absolute timing.

We began by testing how well the phase of a neural oscillator predicts behavioral responses according to relative or absolute timing (see S2A Fig). More precisely we use a Wilson-Cowan model [17], a physiologically inspired neural population model widely used in the literature to explain oscillatory behavior [18–22]. We found that when the oscillator was driven by our ambiguous stimuli its phase was significantly concentrated at the expectations of either relative or absolute perceptual timing, when the mean inter-tone interval of the stimulus maintained a narrow range around the natural frequency of the oscillator. Under these conditions, we found that roving the coupling parameter between the oscillator and the stimulus (see *k* in S2A Fig) modulated the model's alignment between the two perceptual timing methods (see S2B Fig). Furthermore, variation of this coupling parameter recovered the experimentally found

relationship between overall performance and algorithm alignment (i.e., better performance leading to absolute timing alignment, worse performance to relative; see S2D Fig). Despite these promising results, we found that when the trial's mean inter-tone interval varied at the range of the true experiment, no longer matching the natural frequency of the oscillator, the association between its phase and the temporal prediction decreased and the relationship between overall performance and algorithm alignment vanished (see S2D and S2E Fig). We were unable to find a set of parameters for which the Wilson Cowan model could replicate human behavior at the range of temporal intervals used in our experimental task. The oscillatory model clearly lacks key features needed to mimic human behavior (it fails to adjust to new stimulus rates). However, under very restricted conditions, it contains the architecture, required by human performance, to flexibly move between regimes more like relative or absolute perceptual timing algorithms. We further analyzed the model to better understand how the change in coupling parameter led to this shift from relative to absolute timing. S2F Fig shows the phase response curve of the model (i.e., the phase shift in response to a single tone arriving at different parts of the cycle) at varying coupling constants. We show that coupling constants leading to a relative expectation have smaller phase shifts which can incorporate adjustments over many tones, as well as both a stable and unstable synchronization point. For higher coupling constants, on the other hand, each tone produces a large amount of phase shift, independent of which part of the cycle it arrives on, bringing the system more rapidly back to the unique stable point.

Based on this first result, we designed a structure for the oscillator model capable of adapting its natural frequency according to the temporal regularities of the stimulus. One of the parameters of the Wilson-Cowan model represents the input coming from other brain regions. Crucially, the value of this parameter modulates the natural frequency of the system. Given that it is reasonable to assume a dynamic interaction between brain regions, we adopted a model in which the input that the oscillator receives from other brain regions is a function of the perceived auditory period (it is worth noting, that brain activity modulated by the auditory rhythm has been reported in the literature [23]). By assigning dynamics to the input arising from other brain regions, we were able to adjust the natural frequency of the oscillator on each new tone to match the average period of the current trial (see Methods and Fig 3A). Furthermore, we allowed the model to learn across trials allowing it to stay in the space of probable stimulus rates given previous experience.

We found that this modified version of the classic oscillator, the *Adaptive Frequency Oscillator* (AFO), well mimics human behavior. Simulations run with this model (see Methods) showed that the phase of the AFO better predicted early or late according to absolute timing for the 1.2 Hz condition (Wilcoxon signed-rank test, two-sided p < 0.001) and to relative timing for 4 Hz (Wilcoxon signed-rank test, two-sided p = 0.015). No significant difference between timing algorithms was found for 2 Hz (Wilcoxon signed-rank test, two-sided p = 0.81). Also, the AFO yields a similar relationship between timing mechanism alignment and overall performance (Fig 3C) as the experimental dataset. Here, as well as for the behavioral data, we fit the simulations across experimental conditions with polynomial functions of 1, 2, and 3 degrees. We compared the models using the Akaike Information Criterion (AIC) and found the second-order polynomial fit to be best (AIC, first: -86.82, second: -102.87, third: -100.16). This result implies that the oscillator´s phase better predicts early or late as computed by the relative algorithm when overall performance is poor, while for enhanced performance it aligns better to the absolute algorithm estimations.

We also sought to compare this performance to the behavior of a non-oscillatory temporal prediction. For this, we considered a state of the art temporal prediction model used to consider very short sequences, the Sensory Anticipation Module (SAM) from Egger and

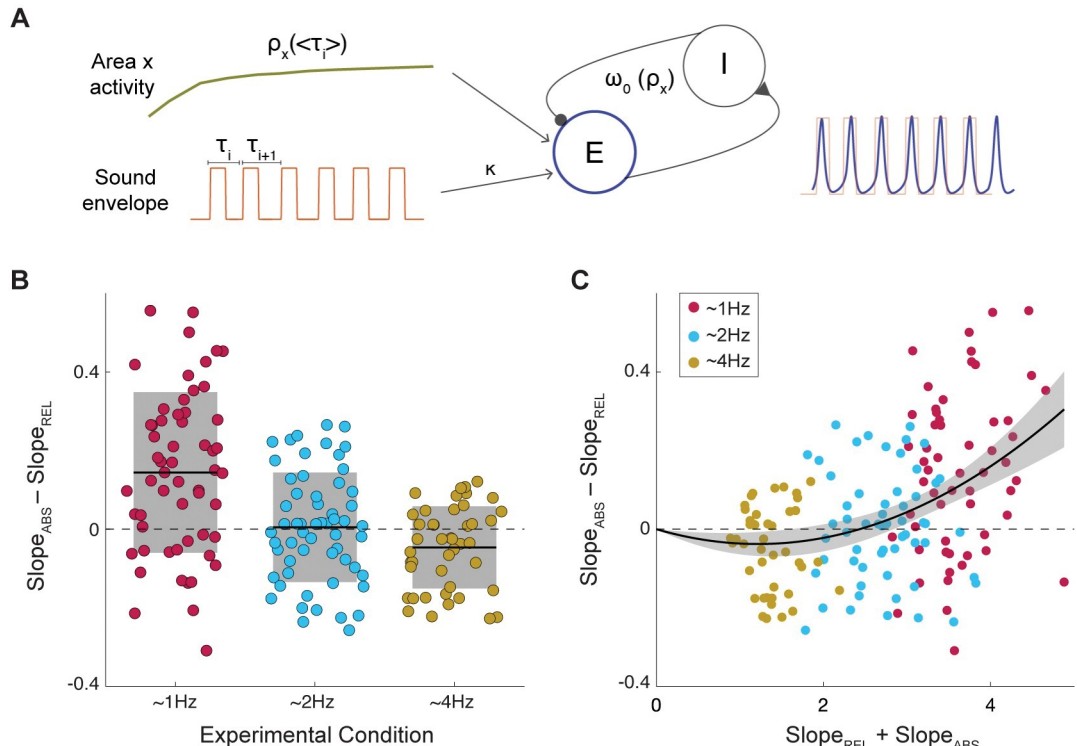

**Fig 3. Adaptive Frequency Oscillator. A.** Schematic of the AFO architecture. Populations of interacting excitatory (E, blue) and inhibitory (I) neurons constitute the Wilson-Cowan oscillator model (see Methods). The sound envelope (orange) drives the excitatory unit by some coupling gain ($\kappa$). The natural frequency of the oscillator is governed by input from an arbitrary area x (green), which stimulates E yielding a faster or slower natural rhythm to match the mean period of the envelope. **B.** Algorithm performance for participants' answers simulated with the AFO in the different experimental conditions ($N_{1Hz} = 60$, $N_{2Hz} = 60$ and $N_{4Hz} = 45$). Difference of the slopes obtained by fitting a logistic regression for the answers computed according to the Duration ($Slope_{ABS}$), or to the Rhythm ($Slope_{REL}$) algorithms as a function of the phase of the oscillator at the time of the probe (see Methods). Black line indicates the mean value and shaded region the standard deviation. **C.** Algorithm alignment as a function of the overall performance. Slope difference between the two logistic regressions as a function of the mean slope across algorithms. The black line represents a second degree polynomial fit and the shaded gray line the 95% Prediction Interval. The parameters of the fitted polynomial are: $y = ax^2 + bx$ with $a = 0.027$ and $b = -0.068$. Simulated data mimic human behavior both in terms of its alignment to relative timing in worse performance and matching the trend of stimulus rates. Data for Fig 3B and 3C can be found in S3 Data.

colleagues [24]. SAM is a predictive ramping model that changes its speed so that the ramping value arrives at a threshold at the predicted expected moment (S3A Fig). For more details about this model please refer to the original paper fully describing it [24]. While simulations with the SAM module yielded high performance, in keeping with its ability on many temporal tasks, it surprisingly failed to replicate the key characteristic behaviors we found on this task. Specifically, while the model's alignment with a given algorithm shifts with stimulus rate, it does not do so monotonically as found in behavior (S2B Fig). More importantly, algorithm alignment with a specific algorithm does not shift with overall performance as found in behavior, instead it is restricted to different frequency groups, yielding a different kind of behavior than what we found in our participants' responses (S3C Fig).

## Bayesian model

Finally, we asked whether the observed behavior reflected efficiency from a computational perspective. In other words, we sought to understand how these results would arise under optimal

conditions, by creating a Bayesian model to simulate the temporal perception of the tones and decision making regarding whether the last tone was early or late. We built a model which represents the timing of each tone as a probability distribution over time. For simplicity, we assume these distributions to be Gaussian. Each tone time is encoded as a gaussian likelihood distribution whose mean is given by the observed timing, $\mu_s$, and whose variance is defined as $\sigma_s$, a value hard-coded for each iteration of the model (representing one "participant"). The measurement is combined with a prior, representing the expectation of where the upcoming tone should be given the previous sequence. The prior is extrapolated from the posterior distributions of previous tones, whose timings and uncertainty are fed to an algorithm to predict the location of the next tone (Fig 4A, left). The algorithm can be used to make expectations either by absolute or by relative perceptual timing, yielding different behavioral results in each case (Fig 4BC). The prior and measurement are combined in a multiplicative fashion to generate the posterior estimate of the true final tone. In the case of the final tone, the prior and likelihood are compared, yielding an estimate of how likely it is that the final tone should be considered early or late. The model's true response is a weighted coin flip based on this probability.

We change only the noise in the sensory measurement to mimic different participants in the experiment. In so doing, we find that the Bayes' model that yields expectations using absolute perceptual timing is remarkably like human behavior we found previously, showing both an agreement with the relative perceptual timing under noisy estimation and an agreement with the absolute perceptual timing in precise sensory estimation. A model using a prior based on relative perceptual timing yields a very different behavior, always aligning with relative timing. These findings lead us to conclude that participants are most likely using an absolute timing-based estimate of their expectations and that the agreement with relative timing in low performance represents evidence of the relative dominance of the prior biasing perception toward isochrony when sensory measurements are noisy.

We next sought to understand how this model would explain the differences that we found across frequencies. We compared two versions of the model: one in which the sensory noise in the model was matched in an absolute sense (Fig 4E) across the stimulus rates, and another in which the noise was matched relative to the mean stimulus rate (Fig 4D), as would be predicted by Weber's Law [25]. The results suggest that differences only arise between relative and absolute timing when the sensory noise is considered to be absolute and not relative to frequency.

Lastly, we sought to test whether the model made any further predictions that could be tested by behavioral data. We found that if one reduces the jitter in time, the model's "relative perceptual timing regime" expands (Fig 4F). We ran a fourth experiment to test this prediction, using the 1.2 Hz range but reducing the jitter in absolute terms to the values used in the 4 Hz experiment. We find that the data confirms the prediction of the model (Fig 4G). Despite better overall performance, the low-jitter experiment yields less absolute perceptual timing dominance overall. We believe therefore that this model yields a straightforward understanding for how temporal prediction occurs under these scenarios.

While the relative strength of the prior depends critically on the strength of the sensory measurement (i.e., the likelihood), it is also capped by the uncertainty inherent to the experiment (see Methods and Materials). As such, as the precision of the likelihood increases, the relative dominance of the prior is reduced (See S4B Fig). The prior, even using the algorithm of absolute timing, assumes equivalent intervals over time, i.e., rhythmicity. As such, when the prior has greater dominance (i.e., when sensory measurements have low precision), the outcome is more aligned with the relative mechanism. When the prior has less influence (i.e., when sensory precision is high and the prior precision is capped), the model is more aligned with the absolute mechanism. This is further confirmed by the predictions of Fig 4F which shows that when the precision cap of the prior is raised (by reducing temporal jitter), the

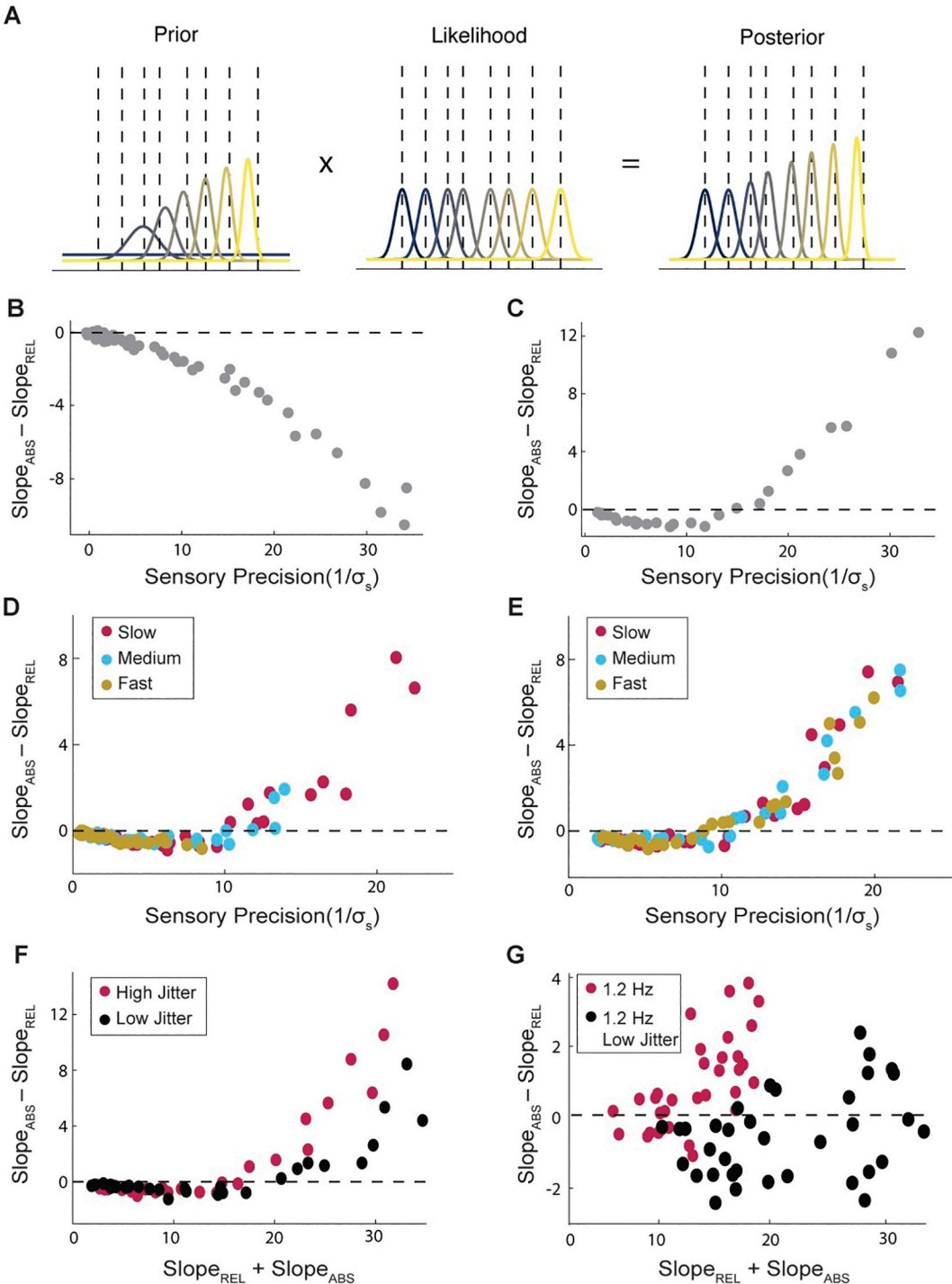

**Fig 4. Bayesian model yields human-like performance. A.** Example trial for temporal estimation of a series of tones in a single trial. The prior (left), likelihood (middle) and posterior (right) of each tone in the sequence. Distributions are color-coded from first to last tone in the sequence from blue to yellow. **BC.** The differences in slope for the model when the prior is estimated using the relative timing algorithm (**B**) or the absolute one (**C**). **DE.** Performance of the model for different stimulus rates (corresponding to our human experiments) where stimulus noise is relative to the stimulus rate (**D**) or the same across rates (**E**). **F.** Performance of the model when the temporal jitter in the tone sequences is altered. Red and black show the high and low jitter conditions, respectively. **G.** Performance of human subjects with lower temporal jitter. Red shows the original 1 Hz condition participants, black shows new participants with reduced jitter. Data for Fig 3B,3C,3D,3E,3F and 3G can be found in S4 Data.

model is aligned with rhythm for a longer period. We confirmed this hypothesis by explaining the difference in temporal jitters shown in Fig 4F (and S4A Fig) by plotting the difference in slopes as a function of the relative ratio between Prior Precision and Sensory Precision in the model. We found that the two levels of temporal jitter become aligned from this viewpoint, confirming that the driving feature in alignment to relative or absolute method of timing is driven by the relative dominance of Prior over Sensation, with alignment for the relative mechanism peaking when Prior and Sensory confidence are near equal (S4C Fig).

## Discussion

Our experiment uses a simple task design to study the added complexity of temporal prediction without perfect rhythmicity. In this setting, we explored participants' responses through the lens of the two classically proposed perceptual timing mechanisms: relative and absolute timing. We found that participants shift their perceptual timing mechanism from relative to absolute, with increasing overall performance. We then built an adaptive frequency oscillator that adjusts its spontaneous rate to match the overall period of the trial. We found that this model mimicked all of the behavior of our human participants, more so than a state-of-the-art predictive ramping model. Our findings push proponents of oscillator models in perception to move beyond basic oscillators to deal with complex stimuli, acknowledging that such units must be placed within the larger neural system. In this case, we placed the oscillator model in an adaptive frequency network, inspired by the adaptive ramp speed of the SAM model that we also tested. In addition, we validate the use of oscillator-based models as a potential means of temporal inference, considering the phase adjustments in response to input as a kind of neural heuristic for a prior expectation for rhythmicity in a Bayesian sense. While Bayesian and dynamical perspectives are not often considered in the same research, we build on recent research suggesting potential for the two to be unified in the temporal domain [26]. Applying this framework in uncertain sequences reveals how oscillatory dynamics can implement expectations for rhythmicity and unify the previous distinction of absolute vs relative timing.

### Human behavior

Our analysis showed that participant behavior varies between relative and absolute perceptual timing. This variation is explained by three major features: the stimulus rate (faster rates were more likely to yield relative timing), overall performance (participants' better overall performance was associated with absolute timing) and the amount of temporal variance in the experiment (less temporal jitter leads to a wider range of alignment to the relative method). When controlling for stimulus variance, overall performance seems to be the dominating feature explaining the timing mechanism alignment, since all stimulus rates can be readily described in terms of the relationship between performance and alignment with a given mechanism using a single parabolic fit. The effect of stimulus rate may be best considered as placing the population on a different part of the function between performance and model alignment. These findings are well explained by our Bayesian model as a set of features which influence the relative dominance of the prior on the likelihood and suggest that strength of the expectation relative to sensory noise in temporal estimates, drives the alignment with the relative method at poor levels of performance.

This single fit across stimulus rates is found in the context of the well-known Weber's law in which overall performance is considered relative to the duration of intervals being perceived. We opted to consider our analysis in this context as is commonly done. Therefore, in this case, the jitter of the probe tone is normalized by the mean tone duration of the

experiment. That normalization yields such alignment across stimulus rates confirms its validity as a means of comparing across frequencies.

## Absolute vs relative timing

Previous work has considered the two perceptual timing mechanisms, absolute and relative, as being supported by two connected timing systems, such that absolute timing is primarily used on irregular sequences and relative timing used on regular/rhythmic sequences. Teki and colleagues [16] developed a unifying theory in which mechanisms associated with relative timing yield a rough temporal estimate based on recently heard intervals and those associated with absolute timing yield error correcting measurements to refine the measurement. By this theory, as irregular sequences have greater error, absolute timing is more heavily relied upon, whereas the reverse is true for regular sequences. While this theory has proved fruitful, we believe it can be refined by considering how the relative timing mechanism yields predictive estimates during irregular sequences. Our modeling results suggest that the two mechanisms might instead be combined not as an error correction process but as the combination of prior and sensory measurement to yield a final temporal estimate. In this sense, the relative timing algorithm may fit as a prior expectation and the absolute mechanism as a likelihood or raw sensory measurement, both estimations combined in a Bayesian sense to yield a final estimation.

Furthermore, from a dynamical perspective, it may be tempting to consider relative timing to relate directly to oscillations and absolute to non-oscillatory mechanisms (e.g., ramping). Our results refute this possibility. We find scenarios in which oscillators, as well as ramping mechanisms, yield either relative or absolute-like behavior. As such, we caution the reader from making this kind of overly simplistic link between human behavior and possible neural mechanisms. In fact, we find that an adaptive frequency oscillator is capable of flexibly navigating the two timing mechanisms representing a unifying theory.

## Biophysical model underlying perceptual timing

While absolute and relative timing are useful frameworks to describe the observed participants' behavior, it is not clear if they actually represent different mechanisms at a neural level (cf. [27,28]). Here, we adopted relative and absolute timing as a useful "coordinate system" to represent participants' responses. Once the experimental outcome was described in this space, we looked for a biologically plausible model capable of explaining the observed results.

We found that an adaptive frequency oscillator (AFO) model well reflected the participants' behavior. This model, in essence, combines elements of other models that often have been used for temporal prediction analysis, in particular oscillator models and predictive ramp models. Here, we found that a simple oscillator could not on its own replicate human behavior in response to ambiguous rhythms, unless the mean rate of the input was near the spontaneous rate of the model. In this restricted scenario, the oscillator could successfully switch between relative and absolute timing depending on its coupling strength, and subsequently overall performance. Meanwhile, the predictive ramp model that we tested, the Sensory Anticipation Module (SAM) designed for use in a tapping study, can successfully make reasonable predictions at a wide array of stimulus rates. Furthermore, it showed differences in model dominance based on stimulus rate. However, overall performance seemed to have no bearing on the model, unlike human behavior. Therefore, while the architecture of SAM is clearly well designed and reflects many aspects of human behavior, it appears to miss something to truly reflect human performance.

Our AFO model is meant as a coarse amalgamation of these two models: using the strengths of each model to ameliorate the weaknesses of the other. AFO contains the oscillatory component which allows for shifting to different models based on performance, and also the adjustment of speed (or period in this case) provided by the architecture of SAM. The combination of the two yields simulations that well resemble human behavior both in terms of behavior across stimulus rates and overall performance, making it a viable candidate for a neural mechanism to underlie temporal prediction. Furthermore, it's important to mention that similar biophysical models have already been used by other authors to explain similar cognitive processes. Specifically, period-adjusting oscillators have been proposed in the music cognition literature as a plausible mechanism underlying beat tracking [29] and general metrical structure perception [30], as well as to explain temporal attention [4]. Similarly, in a recent study Roman and colleagues show that a non-linear oscillator with Hebbian tempo learning and a pulling force toward the spontaneous individual's motor frequency is capable of predicting professional musicians' synchronization on real performance settings [31].

One potential advantage of the AFO is that it also allows for input to adjust frequency not only based on the current stimulus but more abstract information. Added excitatory or inhibitory input from top-down/frontal inputs could theoretically adjust the expected frequency based on contextual knowledge: expectation that the sequence will speed up based on previous tendencies, or the recognition of more complex hierarchical musical rhythms. Such input would plausibly come from motor areas [32]. fit well with recent work by Cannon and Patel [33] which proposes a looping pattern in Supplemental Motor Area (SMA) whose speed is governed by input current. While their proposal is more specific regarding anatomy, we feel that this concept fits well with the AFO framework. Similarly, Rimmele and colleagues [34] proposed oscillatory dynamics in the auditory cortex as a local hard-coded constraint that could be manipulated by top-down control via phase-reset (rather than frequency shift) most likely from cortical motor areas, cerebellum, or basal ganglia.

In addition, another advantage of AFO over the classical oscillator is the reduction of phase precession. When the classical oscillator model is driven by a periodic input, even when it synchronizes, the phase at which the locking takes place is modulated by the stimulus' rate relative to the oscillator's natural frequency [35]. This kind of phase shift is potentially problematic for a model that uses phase as an index of temporal prediction. The AFO postulated in this work overcomes this issue. Given that the natural frequency of the system is adjusted to match the external one, the oscillator synchronizes at zero phase lag for every external rate. Still, AFO does not represent a full account of the neural mechanism, taking several shortcuts to ease computational load and complexity. For example, the model adjusts its own period to match the mean interval of the period without explaining how this mean would be estimated. Future work will determine how this period matching is assessed and in what neural anatomical regions. Still, the key component of our model shows that an oscillator component within a more complex model explains aspects of human behavior that have not yet been explained without it. Such work is in line with advancements in adjacent fields developing the notion of oscillatory behavior into more complex/realistic networks with great success [36–40].

Our AFO model proves that it is plausible that absolute and relative timing can be unified into a dynamical framework. At a neural level there are not two different operations to estimate time, instead, a single neural mechanism can give place to one or another observable behavior.

## Bayesian modeling

To better understand human performance, we developed a Bayesian computational model to see how optimal estimation would occur. We related participants' behavior to inferences in a

Bayesian model and predicted that they should match if the internal model is statistically optimal. Our findings suggest that participant behavior is consistent with Bayesian estimation assuming a prior that expects absolute timing (and not relative). It therefore explains the effect of significant alignment with relative timing at worse performance levels as increasing dominance of the prior in the cue tone phase under scenarios with high sensory noise, which would bias the inference of the model towards equal temporal intervals. When participants are more uncertain about their own sensory measurements, the expectation of similarity between intervals yields a stronger effect of rhythmicity in the estimation.

Interestingly, the Bayesian model makes an added prediction that we had not initially anticipated. As precision in the tone sequence increases, making the sequence closer to isochronous, the relationship between overall performance and timing mechanism alignment flattens, expanding the range in which predicted participants' behavior aligns with relative timing. We ran a fourth experiment using the stimulus rates from the 1.2 Hz experiment but the jitter of the 4 Hz experiment (in absolute terms). Comparing this experiment with the initial 1.2 Hz experiment confirmed the predictions of the model showing a reduction in absolute timing dominance despite higher overall performance. We feel therefore that this model well reflects human behavior and further validates our behavioral findings. It also confirms previous predictions suggested by Teki and colleagues [16] that more regular sequences rely more heavily on relative timing mechanisms.

In showing that both the Bayesian and the AFO model can predict features of human behavior, we explain human temporal predictions in sequence at multiple levels of explanation: from Marr's perspective, computational and algorithmic. This invites the hypothesis that the oscillatory dynamics of the AFO can support Bayesian computation in this kind of task, where participants have expectations for rhythmicity. In this light, the presence of oscillatory dynamics in the prediction of quasi-rhythmic sequences may be best considered as a *hard-coded* expectation for rhythmicity. One that arises not purely from experience from rhythmic sequences, but which is born out of biology: an evolutionary prior that relevant sequences often maintain regular temporal statistics. The notion of hard-coded expectations in time has recently been proposed to explain neural responses to natural delays in audiovisual speech [41]. Here we apply the notion to the rhythmic prediction of sequences themselves. Such a hard-coded expectation for rhythmic stimuli is then fine-tuned by an adaptive signal which adjusts the frequency of the oscillator to predict a specific time scale of rhythmicity. As such, adaptive frequency oscillators represent a neural architecture which balances adaptability and structure in a manner relevant to ecological stimuli.

The Bayesian model adjusts sensory measurements based on how far the prediction is from the sensory measurement, and how confident the prediction is. Meanwhile, the AFO adjusts sensory estimates (in terms of its phase) based on the phase response curve, adjusting the phase of the oscillator depending on the phase at the time of the stimulus, potentially accounting for the distance between prediction and sensory measurement. Prediction confidence, however, is not explicitly accounted for in the AFO model. Future computational work will be required to estimate how confidence affects the neurophysiological model, for example investigating whether oscillation waveform shape [42] or overall amplitude could be an effective neural marker for confidence.

Taken together, our findings show that participants process imprecise rhythms in a manner consistent with the classical definitions of absolute or relative perceptual timing depending on performance (with lower performance for faster rates). Remarkably, we found that an adaptive frequency oscillator well replicated human performance, proving that relative and absolute timing can derive from the same underlying neural mechanism. This model represents a significant advance which we think should guide the development of future oscillatory models of

temporal prediction. In this case, the frequency adaptation represents a mechanism for estimating the period, while the synchronization reflects successful estimation of timing in this period as a kind of neural heuristic for a Bayesian prior. Furthermore, it suggests that frequency adaptation may be a key optimization to the oscillator hypothesis for sensory prediction.

## Materials and methods

### Ethics statement

The study was approved by the University Committee on Activities Involving Human Subjects (UCAIHS) at New York University. All participants provided written informed consent and, as NYU students, were given course credit for their participation.

### Participants

In total, four behavioral experiments were collected (the same task design except for the average tone rate and noise with respect to the underlying isochronous structure): i) 1.2 Hz, ii) 1.2 Hz low jitter (LJ), iii) 2 Hz, and iv) 4 Hz. Across all experiments, we collected data from 215 participants (1.2Hz: 35, 1.2Hz LJ: 37, 2Hz: 78, 4Hz: 65) in a mixture of in person and online experiments (1.2Hz: online, 1.2Hz LJ: online, 2Hz: in person, 4Hz: in person; cf. COVID). Of these, 30 participants were removed as sigmoid fits did not significantly outperform a constant model ($\chi^2$ deviance test). A total of 185 (101 females; mean age, 23 years; age range, 19 to 36 years) participants were analyzed in the study (1.2 Hz: 35, 1.2 Hz LJ: 36, 2 Hz: 63, 4 Hz: 51).

### Experimental design

Participants listened to a series of tones and were asked to identify if the last tone was earlier or later than expected. On each trial, participants hear $N$ tones followed by a final probe tone about which participants give their responses. $N$ is randomly chosen on each trial between 8, 9 or 10 (see Fig 1A). Trials rely on an isochronous underlying structure perturbed according to different parameters. A within-trial average period between tones is selected to define the expected location of each tone. This event period $T$ is drawn from a uniform distribution on each trial whose bounds ($[b_l, b_h]$) were set for each experiment as: 1.2 Hz and 1.2 LJ: [700, 1000] ms, 2 Hz: [400, 600] ms, 4 Hz: [210, 290] ms. Each tone ($t_i$) is then displaced from the expected location by a Gaussian random variable $\varepsilon \sim N(0, \sigma)$ where $\sigma$ is approximately 20% of the mean period across the experiments (1.2 Hz: 170 ms, 2 Hz: 100 ms, 4 Hz: 45 ms), with the exception of the low jitter version of the 1.2 Hz experiment which sought to test the effect of a reduced temporal jitter (1.2 Hz LJ: 45 ms). In all cases, the final probe tone ($t_{probe}$) is placed at the following expected location where the temporal error is drawn from a uniform distribution (see Fig 1A). The tone duration was adjusted for each experiment according to: 0.4 * mean period.

Mathematically, the parameters defining each trial can be defined by the following set of equations, where $U$ represents a uniform distribution, $N$ a Gaussian distribution, $E$ the expectation of a random variable and $I$ the total number of tones within a trial (8, 9, or 10 for any given trial):

$$T \sim U(b_l, b_h)$$

$$t_i = T*i + \epsilon_i \; for \; i = [1, 2, \dots, I]$$

$$\epsilon \sim N(0, \sigma)$$

$$t_{probe} = T*(N+1) + \epsilon_{probe}$$

$$\epsilon_{probe} \sim U(-.3*E[T], .3*E[T])$$

Each experiment comprised 150 trials and each participant was assigned to one of the 4 experiments: 1.2 Hz LJ, 1.2 Hz, 2 Hz or 4 Hz.

To familiarize participants with the task, we implemented a brief training period to ensure that the task was clear and sensical. Participants first heard one trial where the tones were perfectly rhythmic, and the probe tone was substantially deviated from the expected time. Then, participants heard 8 more trials where tone timing was jittered but with variance equal to one third of the experimental variance. The probe timing was again significantly deviated such that the correct answer by the two absolute and relative timing expectations would always align. Participants received feedback on each trial of their performance. Finally, the participants were given 5 more trials with timing variance at one half of experimental variance without feedback. After this point, the researcher verified that the participant understood the task and felt clear on their goals before continuing to the main experiment described above.

## Experimental analysis

**i. Absolute vs relative timing frames.** While the tones are generated by a specific set of equations, the listener does not have access to this generating algorithm and must therefore infer it based on very limited data (10 tones at most). In fact, the task can be solved by different "algorithms" leading to different results as to whether the final tone is early or late. We considered two possible algorithms in particular that align with the two typically proposed perceptual timing mechanisms [16,43]: absolute (or duration-based) and relative (or rhythm-based). They are exemplified in Fig 1AB. The *relative* algorithm assumes that participants use the sensory evidence only to infer where the expected (mean) location would have been, while the *absolute* algorithm takes sensory evidence at face value and adds the mean interval on to the timing of the previous tone. The main difference between these algorithms is whether you should allow for drift in the expected locations. In that sense, for the relative timing algorithm the time of the incoming tone is predicted by computing the overall temporal regularity in the preceding sequence. In contrast, for the absolute timing algorithm it is the absolute time that goes by between tones that drives prediction.

A primary question for our study was to uncover, given a tone sequence, how do participants navigate this distinction and deal with the lack of a ground truth to generate a response. To answer this, we score participant data by *both* algorithms to help us identify which one the participants behavior most closely resembles. To do so, probe tones are coded by their deviation from either the relative prediction or the absolute algorithm prediction. Then, we fit two sigmoid functions between this deviation (from the two algorithms) and the response of the participant as to whether the tone is early (recorded as 0) or late (1) as in Fig 1C. We then extract the slope parameter of this fit as a measure of the precision of their performance for either algorithm or compare the two obtained slopes to see which algorithm performs best in terms of sorting the participants' responses (i.e., which fitting shows a steeper slope). Fig 1C shows an example of a subject who performs *best* according to the absolute algorithm. For analyses merging the data of different experiments (i.e., different rates) the normalized logistic regression was estimated. Meaning that the time difference between the probe tones and the absolute and relative algorithms' predictions were divided by the mean stimulus onset asynchrony (SOA) of the corresponding trial.

Logistic regressions on each participant's responses were fitted using the fitglm matlab function, which returns a generalized linear model fit. Participants with a noisy response pattern ($\chi^2$ statistics against constant model, p>0.05) were excluded from the follow up analyses. The number of participants excluded for each experiment are: 14 from a total of 65 for the 4 Hz experiment, 15 from a total of 78 for the 2 Hz, and 1 from a total of 72 for the 1.2 Hz experiments (including both 1.2 Hz and 1.2 LJ).

**ii. Statistics.**   All between- and within-subject comparisons were assessed by nonparametric Mann-Whitney-Wilcoxon and Wilcoxon signed-rank tests, respectively.

The precision of absolute and relative algorithms were first plotted against each other. Then, to optimize analysis, the axes were rotated to highlight algorithm dominance (absolute—relative) and overall precision (absolute + relative). We then ran polynomial fits on this rotated version to estimate regions of absolute and relative algorithm dominance. We fit polynomials from first to third order and compared the fits using Akaike Information Criterion (AIC) to estimate how overall performance relates to method behavior. The model is fitted using a least-squares estimation using the matlab fitglm function. 95% confidence intervals of the model predictions were extracted from the model fit and compared against 0 to assess when the function is significantly above or below 0. The overall goodness of the fit was estimated by means of the adjusted R-square, penalizing the variables addition to the model.

## Computational modeling

We designed two model types to explain the behavioral effect in terms of: i) neural dynamics, and ii) computational processes and efficiency. First, we designed an adaptive frequency oscillator as a candidate neural mechanism to underlie the experimental pattern. Next, we used a Bayesian approach to understand how the combination of expectation and sensory measurement might yield similar human behavior.

**i. Adaptive frequency oscillator.**   The Wilson-Cowan (WC) model is a biophysically inspired neural mass model which has been widely used in the literature and shows a rich set of possible dynamics [17,44]. This model assumes that a given brain region is composed by an excitatory and an inhibitory neural population interacting with each other and can be described by the following set of equations:

$$\tau \frac{dE}{dt} = -E + S(\rho_E + cE - aI + k\ Stim)$$

$$\tau \frac{dI}{dt} = -I + S(\rho_I + bE - gI)$$

Where $E$ represents the excitatory population, $I$ the inhibitory one, $S$ is a sigmoid function, $a,b,c$ and $g$ represent the synaptic coupling, $\tau$ is the membrane time constant, $Stim$ is the external stimulus driving the brain region activity, $k$ is the strength of the coupling between the brain and external stimulus, and $\rho_E$ and $\rho_I$ are stable inputs that the different populations receive from distant brain areas. In the current work, a model like this has been adopted to represent auditory regions, and the external signal driving the excitatory population (i.e., $Stim$) has been assumed to be proportional to the broadband envelope of the auditory stimulus:

$$Stim = \frac{Audio\ Envelope - <AudioEnvelope>}{max(Audio\ Envelope - <AudioEnvelope>)}$$

$$Audio\ Envelope = |hilbert(Audio\ Signal)|$$

Importantly, the set of equations defining the WC model simulate different bifurcations for different combinations of $(\rho_E, \rho_I)$. In previous work [13,45], where a neural oscillator is hypothesized, the parameters $(\rho_E, \rho_I)$ were fixed and set close to an Andronov-Hopf bifurcation where the system behaves as an oscillator with a fixed natural frequency. We refer to this selection of parameters as the classic oscillator, and simulations for S2 Fig were computed with this model. To get an adaptive frequency oscillator, we make use of a SNIC Bifurcation (Saddle Node on an Invariant Circle) allowing for the adjustment of parameters $(\rho_E, \rho_I)$ to modify the internal natural frequency. We assigned them the following dynamics:

$$\rho_I = -7$$

$$\frac{d\rho_E}{dt} = -0.045(\rho_E - \rho_0)$$

$$\rho_0 = -3.4\ during\ silence$$

$$\rho_0 = \frac{0.45 < t_i >}{< t_i > -0.21} - 3.6\ during\ sound$$

Where $t$ represents the period of the perceived sound (i.e., the time interval between sounds is estimated and the mean value is actualized right after each tone). To select the functional form for $\rho_0$, we numerically explored the relationship between the natural period of the system and $\rho_E$ (see Fig 5). The functional form for $\rho_0$ has been chosen to drive the system to match the

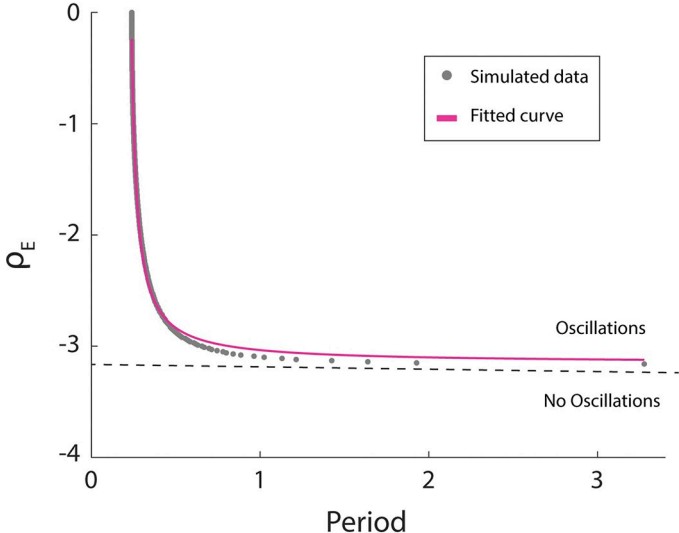

**Fig 5. Oscillatory behavior of the WC model.** Running numerical simulations of the WC equations with no stimulus (*Stim = 0*), we estimated the period of the excitatory population activity for different values of $\rho_E$ (gray dots). A saddle node in limit cycle bifurcation takes place around $\rho_0 = -3.16$ (dashed line). Such a bifurcation gives birth to very slow oscillations rapidly increasing its frequency as the relevant parameter departs from the bifurcation point. We fitted a rational function to the numerical data to get an analytic parametrization of $\rho_E$ as a function of the natural period of the system. Blue traces depict the activity of the excitatory population in the different regimes. Data for Fig 5 can be found in S5 Data.

perceived auditory period. The chosen dynamics places the system during silence at rest ($\rho_E$ = −3.4) and close to a saddle node in limit cycle bifurcation. When the sequence of tones begins $\rho_E$ evolves in time through $\rho_0$, crossing the bifurcation and allowing the system to adjust the natural frequency of the oscillator to match the rhythm of the stimulus.

## Simulated behavior

To simulate behavior with the AFO the parameters of the model were set to: a = b = c = 10, d = -2 as typically selected in the literature [44], **$\tau$ = 1/17** and (**$\rho E$, $\rho I$**) as stated in the previous section. Different participants were modeled by varying the coupling value as: **$k$ = 0.2+$U$(0, 0.1)** and 300 different auditory trials were evaluated per participant. The time interval between trials was assigned randomly as: **$ITI$ = 0.75+$U$(−0.25, 0.25)** and stimuli were generated in the same way as for the behavioral protocol but removing the probe tone. For each trial the phase of the AFO at the time where the probe should appear was estimated as the phase of the Hilbert transform of the excitatory population activity. The AFO performance was computed by fitting a logistic regression to adjust early or late responses (0 or 1) according to relative and absolute timing algorithms as a function of the oscillators phase (using (**$cos(\theta)$, $sin(\theta)$**)), for each simulated participant (i.e., for each set of 300 phases computed as stated before). As well as for the experimental data, simulated participants with a noisy response pattern ($\chi^2$ statistics against constant model, p>0.05) were excluded. We simulated a total of 60 participants per experimental condition, and 15 simulated participants from the 4Hz experimental condition were excluded due to noise response pattern.

In S2 Fig, in addition to the above simulation, we also use phase concentration to assess what information is contained within the classical oscillator (i.e., the Wilson-Cowan Model with no dynamics assigned to $\rho_E$). We use the Hilbert transform to estimate phase of the excitatory population activity at the *expected* time as predicted either by relative or absolute timing across 300 different trials and use phase concentration to estimate how consistent this phase is across trials. Higher concentration infers that the phase is synchronized to this expected timing. Phase concentration is assessed using the following equation:

$$PC_X = \frac{|\sum_{i=1}^{T} e^{i\theta_X}|}{T}$$

All simulations were run in MATLAB using a standard Euler solver with a time step of 0.0001 sec.

The Phase Response Curve in S2F Fig was analyzed by freezing the oscillator start point and running it for four cycles (1 s) as a reference. Then we simulated the Wilson-Cowan model again with the same start point but adding a tone stimulus identical to the one used in task simulation but roving its onset from the beginning to the end of the second cycle using 1 ms step size. Then we averaged the phase difference between perturbed and reference oscillators in the fourth cycle. This phase difference as a function of phase of stimulus onset is plotted S2F Fig at coupling constants of 0.02, 0.1, 0.2, 0.3, 0.4, 0.5, 0.6, 0.7, 0.8, 0.9, and 1.0.

**ii. Bayesian model.** This model, as Bayesian, is founded on the principle that a sensory measurement, a likelihood, is combined with an expectation, or prior, to yield a final estimate of each tone's timing, a posterior distribution.

$$N(t|\mu_{e,i}, \sigma_{e,i}) \propto N(t|\mu_{p,i}, \sigma_{p,i}) * N(t|\mu_{s,i}, \sigma_{s,i})$$

Where $\mu_{e,i}$ and $\sigma_{e,i}$, refer to mean and standard deviation of the posterior distribution, the final estimate, of the i-th tone; $\mu_{p,i}$ and $\sigma_{p,i}$, refer to mean and standard deviation of the prior distribution of the i-th tone and $\mu_{s,i}$ and $\sigma_{s,i}$, refer to mean and standard deviation of the

likelihood distribution, the model sensation, of the i-th tone. As we have chosen to model these distributions as Gaussian, $\mu_{e,i}$ and $\sigma_{e,i}$ can be calculated as follows:

$$\mu_{e,i} = \frac{\frac{\mu_{p,i}}{\sigma_{p,i}{}^2} + \frac{\mu_{s,i}}{\sigma_{s,i}{}^2}}{\frac{1}{\sigma_{p,i}{}^2} + \frac{1}{\sigma_{s,i}{}^2}}$$

$$\sigma_{e,i} = \sqrt{\frac{1}{\frac{1}{\sigma_{p,i}{}^2} + \frac{1}{\sigma_{s,i}{}^2}}}$$

Estimates of the stimulus timing of an event, the likelihood are measured with gaussian noise, such that if the i-th tone occurs at time $t_i$, then the model's measurement of this time will be a gaussian distribution $N(\mu_{s,i}, \sigma_{s,i})$ where $\sigma_{s,i}$, the standard deviation of the gaussian, represents sensory noise or inversely the precision of the model representation and where $\mu_{s,i} = t_i$. We have elected to use a separate term $\mu_s$ to refer to the model's mean sensation of time as while it is numerically the same as $t_i$, conceptually it refers to an internal parameter of the model rather than the external true value of the tone's timing. Next, we assume that this noisy estimate of time is combined with a prior of the tone's expected timing based on preceding tones. We devise this prior in two possible methods following the same relative and absolute timing algorithms we used in the behavioral analysis.

The relative timing prior uses linear regression with unequal variance to estimate the mean and variance of the next expected location. Thus, the distribution representing the estimate and uncertainty of the expected timing of the next tone can be calculated with the following equation assuming t = 0 is defined as the start of the first tone.

$$\mu_{p,i} = \frac{i * \sum_{n=1}^{i-1} \left( \frac{i * \mu_{e,n}}{\sigma_i{}^2} \right)}{\sum_{n=1}^{i-1} \left( \frac{i^2}{\sigma_{e,n}{}^2} \right)}$$

$$\sigma_{p,i}{}^2 = \frac{i^2}{\sum_{n=1}^{i-1} \left( \frac{i^2}{\sigma_{e,n}{}^2} \right)} + \sigma_{exp}{}^2$$

Where $i$, is the index of the tone for which the prior is to be estimated, $n$ is the index of all preceding tones up to $i-1$, and $\sigma_{exp}$ is the uncertainty due to the experiment, the standard deviation of the jitter applied to each tone which we assume the participant knows *a priori*.

The absolute timing prior calculates the difference between neighboring tone times to estimate the mean interval between tones and adds this interval to the final tone time to generate an estimate of the prior. The mean and variance are calculated as follows:

$$\mu_{p,i} = \frac{\sum_{n=1}^{i-2} (\mu_{e,n+1} - \mu_{e,n})}{i - 2} + \mu_{e,i-1}$$

$$\sigma_{p,i}{}^2 = \frac{\sum_{n=1}^{i-2} (\sigma_{e,n+1}{}^2 + \sigma_{e,n}{}^2) - 2\sum_{n=2}^{i-2} (\sigma_{e,n}{}^2)}{(i-2)^2} + \sigma_{e,i-1}{}^2 + 2\frac{\sigma_{e,i-1}{}^2}{i-2} + \sigma_{exp}{}^2$$

Where the terms $-2\sum_{i=2}^{i-2} (\sigma_{e,n}{}^2)$ and $2\frac{\sigma_{e,i-1}{}^2}{i-2}$ account for the covariance between neighboring intervals and between the mean interval and the final tone time, respectively.

Both priors, the relative and the absolute timing, apply only after the first two tones have been presented. For the first two tones, we assume a flat prior such that the posterior

distribution is equal to the sensory measurement, $N(t|\mu_s, \sigma_s)$ This model setup yields a posterior distribution for the estimate of each tone within the sequence which is then applied to the generation of priors for upcoming tones. Note that the posterior for each tone is based purely on preceding tones and is not updated based on later information.

Finally, the model must decide as to whether the final tone is earlier or later than expected. In this case, rather than combining prior expectation and sensory measurement, we compare the two distributions to assess the probability that the sensory estimate is higher than the prior expectation by marginalizing over time the product of cumulative distribution of the prior and probability distribution of the sensory estimate.

$$P(late) = \int_{-\infty}^{\infty} \Phi_{\mu_p, \sigma_p}(t) * N(t|\mu_s, \sigma_s) dt$$

$$\Phi_{\mu, \sigma}(x) \equiv \int_{-\infty}^{x} \frac{exp\left(-\frac{(x-\mu)^2}{2\sigma^2}\right)}{\sqrt{2\pi\sigma^2}} dx$$

Using this probability, $P(late)$, we flip a weighted coin based on the probability distribution to determine if the model responds late or early to the trial, and analyze the output exactly as described in the data analysis section on human behavior. We then investigate the model's behavior by roving the amount of noise in the sensory estimate $\sigma_s$ to see how the model behaves with sensory noise.

## Supporting information

**S1 Fig. Trials design and different predictions of the two computational mechanisms of perceptual timing. A.** Relationship between the amount of noise in the last tone of the sequence (how much it departs from the underlying rhythmic structure) and the difference between the probe's time predicted by the two models. **B.** Relationship between the sum of the noises in the sequence of tones preceding the last one and the difference between the probe's time predicted by the two models. Panels **A&B** imply that the more the trials depart from the perfect isochronous case, the larger the difference between the models predictions. Additionally, accordingly to how the jitter accumulates on the different models, the noise in the last tone has the strongest effect in differentiating both predictions. Each dot represents one trial and the solid line a linear regression of the data. **C.** Schematic representation of four trials of the 4Hz rhythm condition. Dashed gray lines: Underlying rhythmic structure (i.e., where the tones of the sequence should be presented for a perfect isochronous stimulus). Solid gray lines: Presented tones. Green: Probe tone. Dashed blue and magenta line: Probe's predicted time according to the relative and absolute perceptual timing, respectively. Congruent trial: both models agree on the probe being early or late. Incongruent trial: there is a disagreement between the models categorical outcome (i.e., probe happens early or late). As exemplified by this figure, the different models' categorical outcome not only depend on the trail structure but also on the timing of the probe tone. While it's true that the larger the difference between the models predictions the more likely is to get an incongruent trial and vice versa (i.e., Trials 1 and 2), it's also possible to find trials with: 1. a small difference between models but with the probe taking place in the middle granting an incongruent trial (i.e., early or late, Trial 3); or 2. a large difference between models but both lying on the same side of the probe granting a congruent trial (i.e., early or late, Trial 4). There is no strict functional relationship between the sequence of tones structure and getting a congruent or an incongruent trial. Data for S1 Fig can be found in S6 Data.
(PNG)

**S2 Fig. Classic oscillator simulations. A**. Set up of a Wilson-Cowan oscillator model (see Methods) with parameters set at: a = b = c = 10, d = -2 and (E,I) = (1.6, -2.9). The acoustic envelope of a stimulus trial drives the excitatory population with coupling determined by parameter k. The phase of the oscillator at the expected time of the last tone according to absolute timing (ABS) or the relative timing algorithm (REL) is computed on each trial. **B**. Phase concentration of predicted phases, ABS in purple and REL in blue across trials at a restricted range of stimulus rates (240 to 260 ms). Better phase concentration would lead to a more accurate prediction of the probe time relative to the corresponding perceptual timing mechanism. Shaded areas mark significant differences using the circular K test to test for significant differences in concentration (correcting for multiple comparisons using the false discovery rate Benjamini & Hochberg, 1995). Insets represent example concentrations at = 0.15, left, and = 2.0, right. **C**. Same as b but with a range of stimulus rates that reflects the statistics of the experiment (210 to 290 ms). **D**. Model task performance compared between Absolute and Relative algorithms in the restricted range of stimulus rates (240 to 260 ms). The difference in slope parameters fitting a Logistic regression between phase of the oscillator at probe time and the correct response defined either by absolute or relative perceptual timing mechanisms (see Methods). Polynomial fit and confidence shown in black line and gray patch respectively. Second order determined through AIC Model selection. **E**. Same as D with the broader range of stimulus rates (210 to 290 ms). **F**. Phase Response Curve of the Wilson-Cowan oscillator in response to a single 100 ms tone at a range of coupling parameters (from 0.02 to 1). Blue lines refer to coupling constants that lead to significantly higher concentration for the relative prediction, Purple to those significantly higher for the absolute one and gray to those with no significant difference between them. Phase response curves consistent with relative timing show a topologically distinct behavior than the other two categories. Data for S2 Fig can be found in S7 Data.
(PNG)

**S3 Fig. Performance of the Sensory Anticipation Module. A.** Model schematic of the ramping model (adapted from Egger and colleagues[23]). The model contains two competing units us and vs whose values decay to a stable point, driven by current I controlling the speed of this decay. Their difference yields $y_s$, a ramping value which is compared with threshold, $y_0$, at the time of a stimulus. The difference $d = y_s - y_0$ at the time of each tone is used to adjust I controlling the speed of the ramp to reduce d on the next interval. In our case, d is also used at the time of the probe tone to output a response to the behavioral trial. If $d > 0$, the model responds "late"; if $d < 0$, the model responds "early". For code and further description of the module, see Eggers et al, 2020. **B.** The model responses are then treated as behavioral data. Late and early responses are coded as 1 and 0 respectively and a logistic function is fitted, and the slope extracted to identify the precision of the responses relative to the relative and duration algorithms. Same procedure as the one applied to the behavioral data. Slope differences are shown here by stimulus rates at 1.2 Hz (red), 2 Hz (blue) and 4 Hz (gold). **C.** Slope difference relative to overall performance (slope sum) for the same stimulus rates. Black line represents polynomial fit, while the gray area represents the 95% confidence interval of the mean. Third order polynomial selected through AIC model selection. Data for S3 Fig can be found in S8 Data.
(PNG)

**S4 Fig. Reducing temporal jitter increases relative alignment by strengthening the prior. A.** Bayesian simulation of the low jitter (black) and high jitter (magenta) conditions. Equivalent to Fig 4F. **B.** Comparison of the Sensory Precision $\frac{1}{\sigma_s}$ with the Prior Precision $\frac{1}{\sigma_p}$ in the low jitter (black, solid line) and high jitter (magenta, solid line) conditions. Horizontal dashed

lines refer to $\frac{1}{\sigma}$ for each experimental jitter: $\sigma = .1$ for high jitter (magenta) and $\sigma = .075$ for low jitter (black). These are the limit of possible precision of the prior if it were 100% confident where the expected tone location would be (considering the unpredictable variance of the experiment). Gray line dashed line refers to the identity line. **C.** The low and high jitter conditions are fully aligned when explained by sensory precision relative to the prior $\frac{\sigma_p}{\sigma_s}$. Data for S4 Fig can be found in S9 Data.
(PNG)

**S1 Data. Data for Fig 1C.**
(XLSX)

**S2 Data. Data for Fig 2A,2B and 2C.**
(XLSX)

**S3 Data. Data for Fig 3B and 3C.**
(XLSX)

**S4 Data. Data for Fig 4B, 4C, 4D, 4E, 4F & 4G.**
(XLSX)

**S5 Data. Data for Fig 5.**
(XLSX)

**S6 Data. Data for S1 Fig.**
(XLSX)

**S7 Data. Data for S2 Fig.**
(XLSX)

**S8 Data. Data for S3 Fig.**
(XLSX)

**S9 Data. Data for S4 Fig.**
(XLSX)

## Acknowledgments

The authors would like to thank David Poeppel, Joan Orpella for their comments and helpful discussion.

## Author Contributions

**Conceptualization:** Keith B. Doelling, M. Florencia Assaneo.

**Data curation:** Keith B. Doelling, M. Florencia Assaneo.

**Formal analysis:** Keith B. Doelling, M. Florencia Assaneo.

**Funding acquisition:** Keith B. Doelling, Luc H. Arnal, M. Florencia Assaneo.

**Investigation:** Keith B. Doelling, Luc H. Arnal, M. Florencia Assaneo.

**Methodology:** Keith B. Doelling, M. Florencia Assaneo.

**Project administration:** Keith B. Doelling, Luc H. Arnal, M. Florencia Assaneo.

**Resources:** Keith B. Doelling, M. Florencia Assaneo.

**Software:** Keith B. Doelling, M. Florencia Assaneo.

**Supervision:** Luc H. Arnal, M. Florencia Assaneo.

**Validation:** Keith B. Doelling, M. Florencia Assaneo.

**Visualization:** Keith B. Doelling, M. Florencia Assaneo.

**Writing – original draft:** Keith B. Doelling, M. Florencia Assaneo.

**Writing – review & editing:** Keith B. Doelling, Luc H. Arnal, M. Florencia Assaneo.

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
