## [Decision Letter · Decision Letter 0]

20 Aug 2023

Dear Dr Assaneo,

Thank you very much for submitting your manuscript "Adaptive oscillators support Bayesian prediction in temporal processing" for consideration at PLOS Computational Biology. As with all papers reviewed by the journal, your manuscript was reviewed by members of the editorial board and by several independent reviewers. The reviewers appreciated the attention to an important topic. Based on the reviews, we are likely to accept this manuscript for publication, providing that you modify the manuscript according to the review recommendations.

Sincerely,

Lyle J. Graham

Section Editor

PLOS Computational Biology

Reviewer's Responses to Questions

**Comments to the Authors:**

Reviewer #1: This paper is an excellent combination of creative experimental work and modeling at multiple levels. It certainly deserves publication in PLOS Comp Bio. My only concerns regard the manner in which the ideas are communicated. There are several points on which clearer communication or some rethinking of underlying ideas would be valuable. There is also a bit of a hole in the fabric of citations holding up the work: previous examples of adaptive oscillators and behavioral models of early/late timing judgement have been largely omitted. Hopefully we can get these minor issues out of the way and get this thing published ASAP.

Main concerns:

- Throughout, overall participant performance is treated as an independent variable (or, even stranger, as a "mediating variable" between frequency and preferred timing strategy). The authors say that overall performance can "explain" variation in timing algorithm. This choice may make for a clear story, but represents a very strange way of thinking about task performance. The graph relating overall performance with algorithm preference makes sense, but it represents a *relation*, not a function. There must be one or more variables, causally "upstream" of task performance, that parametrize this curve, and that may be affected by frequency. I would love to see the language change throughout to represent this.

- Patchy citation coverage: throughout, there is a conspicuous lack of mention of preceding models of period-adjusting oscillators. Jones (and Large, and Palmer, etc.) were already working with this type of model in the late 90's. They were even applied to model interval timing tasks: when the authors discuss using oscillators for "absolute timing" (Absolute vs Relative Timing section), they should mention McAuley, J. D., & Jones, M. R. (2003). Modeling Effects of Rhythmic Context on Perceived Duration: A Comparison of Interval and Entrainment Approaches to Short-Interval Timing. Journal of Experimental Psychology: Human Perception and Performance, 29(6), 1102–1125. https://doi.org/10.1037/0096-1523.29.6.1102. Also, Li et al, "For the last time: temporal sensitivity and perceived timing of the final stimulus in an isochronous sequence" does a nice job of reviewing and evaluating possible algorithms for early/late tasks -- definitely worth a read and a citation, and might help to collect some other citations.

- There are at least three distinct uses of the word "prior" -- the time at which an event is initially expected, the rules (absolute vs. relative) determining the position and spread of that prior, the hard-coded belief that stimuli tend to be rhythmic... This is especially confusing in the first paragraph under "Bayesian Modeling." Phrases like "a prior adjusting timing toward rhythmicity" are confusing since it sounds like the rules determining the priors are changing even though they are not. Overall, some more careful wording around these priors would be helpful to disambiguate between the rules determining the priors, the priors themselves, and the emergent timing algorithms (absolute vs. relative) that act as poles of the space of possible participant strategies.

Fairly minor:

- This task is a bit unusual in that participants are being asked to make a judgement that does not correspond with a particular reality in the world, only with their own intuitive sense of expectation. Can you offer any reassurance that participants found this task sensical? (Orderly data is a good sanity check, but is there anything else to go on?)

- "As an alternative, we adopt a data driven approach, where the two classical timing computational mechanisms are determined by the participants' responses." Fuzzy language. How about, "where the two classical timing computational mechanisms emerge as opposite poles of participant behavior that ranges from one to the other."

- Participants in the 2 Hz condition showed no significant preference, p=0.022. I'm probably missing something, but isn't that "significant"? Are you using a threshold below 0.05?

- "significant period at worse overall performance." "period" is a vague word -- choose something clearer?

- "under certain parameter spaces" -- similarly vague

- "Algorithm prevalence" -- for consistency, shouldn't this be "preference"?

- "Neural precision" -- was this term explained? Is there a better term for this, or a term that lines up with the math notation? I don't even see where eta is defined.

- Fig. 4 F/G -- why not use colors coordinated between F and G?

- "reduces jitter in time between participants" -- what about this is "between participants"?

- "the range in which relative timing performs best" -- shouldn't this be "the range in which participants seem to be using a more relative timing strategy"?

- "...predictions of the model. Showing a reduction..." -- misplaced period.

- "Marrian" -- I'm not a fan of this adjectivication of "Marr."

- the "d" parameter in the AFO leads to two possible meanings of "dI" -- awkward notation, though I guess it's standard.

Reviewer #2: Attached.

**Have the authors made all data and (if applicable) computational code underlying the findings in their manuscript fully available?**

Reviewer #1: **No: **available "upon request."

Reviewer #2: Yes

PLOS authors have the option to publish the peer review history of their article (what does this mean?). If published, this will include your full peer review and any attached files.

Reviewer #1: **Yes: **Jonathan Cannon

Reviewer #2: **Yes: **Benjamin Pittman-Polletta

Figure Files:

Data Requirements:

Reproducibility:

References:

---

## [Decision Letter · Decision Letter 1]

7 Nov 2023

Dear Dr Assaneo,

We are pleased to inform you that your manuscript 'Adaptive oscillators support Bayesian prediction in temporal processing' has been provisionally accepted for publication in PLOS Computational Biology.

Before your manuscript can be formally accepted you will need to complete some formatting changes, which you will receive in a follow up email. A member of our team will be in touch with a set of requests. I also ask that you take a look at the suggestions of Reviewer 2, which seem quite reasonable.

Best regards,

Lyle J. Graham

Section Editor

PLOS Computational Biology

Reviewer's Responses to Questions

**Comments to the Authors:**

Reviewer #1: This revision is totally satisfactory. The language has been clarified and errors have been fixed.

Reviewer #2: The authors have done a great job addressing both reviewers’ comments. I think the manuscript is significantly clarified and improved. I only have a few small comments:

• Fig. 1A – it is hard to tell the green dot is green.

• Line 338: “While the strength of the prior depends critically on the strength of the sensory measurement” – “strength” should be “relative strength”.

• Line 344: “when sensory measurements are high” – should be “when sensory precision is high”.

• Line 349: “by explaining the difference in slopes” – should be “by plotting the difference in slopes”.

• Line 422-3: sentence fragment, “This ability to align with relative”.

• Unless I’m mistaken, it is still not clear that the coupling parameter of the AFO is varied to fit each participant’s data to get the results shown in Fig. 4. Perhaps this could be mentioned somewhere around line 260?

• Unfortunately, the equations did not show up in the revised manuscript. This should be addressed. I’m willing to take it somewhat on faith that the equations have been sufficiently clarified; perhaps the authors can make a special effort to proofread them before publication.

**Have the authors made all data and (if applicable) computational code underlying the findings in their manuscript fully available?**

Reviewer #1: **No: **The authors say that code is available on request.

Reviewer #2: Yes

PLOS authors have the option to publish the peer review history of their article (what does this mean?). If published, this will include your full peer review and any attached files.

Reviewer #1: **Yes: **Jonathan Cannon

Reviewer #2: **Yes: **Benjamin R. Pittman-Polletta

---

## [Editor Report · Acceptance letter]

17 Nov 2023

PCOMPBIOL-D-23-01075R1 

Adaptive oscillators support Bayesian prediction in temporal processing

Dear Dr Assaneo,

I am pleased to inform you that your manuscript has been formally accepted for publication in PLOS Computational Biology. Your manuscript is now with our production department and you will be notified of the publication date in due course.

With kind regards,

Zsofi Zombor
